# Motion Diffusion-Guided 3D Global HMR from a Dynamic Camera

## Abstract

Motion capture technologies have transformed numerous fields, from the film and gaming industries to sports science and healthcare, by providing a tool to capture and analyze human movement in great detail. The holy grail in the topic of monocular global human mesh and motion reconstruction (GHMR) is to achieve accuracy on par with traditional multi-view capture on any monocular videos captured with a dynamic camera, in-the-wild. This is a challenging task as the monocular input has inherent depth ambiguity, and the moving camera adds additional complexity as the rendered human motion is now a product of both human and camera movement. Not accounting for this confusion, existing GHMR methods often output motions that are unrealistic, e.g. unaccounted root translation of the human causes foot sliding. We present **DiffOpt**, a novel 3D global HMR method using **Diff**usion **Opt**imization. Our key insight is that recent advances in human motion generation, such as the motion diffusion model (MDM), contain a strong prior of coherent human motion. The core of our method is to optimize the initial motion reconstruction using the MDM prior. This step can lead to more globally coherent human motion. Our optimization jointly optimizes the motion prior loss and reprojection loss to correctly disentangle the human and camera motions. We validate **DiffOpt** with video sequences from the Electromagnetic Database of Global 3D Human Pose and Shape in the Wild (EMDB) and Egobody, and demonstrate superior global human motion recovery capability over other state-of-the-art global HMR methods most prominently in long video settings.

## 1 Introduction

3D human mesh recovery (HMR) refers to the task of computing a mesh of a human body in 3D given an input image or a video. HMR has various applications such as augmented/virtual reality, motion capture (MoCap), sports, and healthcare. Particularly, in terms of MoCap, HMR holds the advantage in terms of accessibility and cost over traditional marker-based MoCap systems that require costly equipments and human subjects to wear specialized marker suits. In light of this demand for more accessible methods of MoCap, numerous optimization-based HMR algorithms have been developed in recent times Mündermann et al. (2006); Nagymáté & Kiss (2018); Hamill et al. (2021); Colyer et al. (2018). Moreover, HMR methods that not only predict the pose of the human body but also the global root trajectory have garnered significant attention. We refer to this task as global HMR (GHMR). Though several GHMR algorithms have been developed recently Yuan et al. (2022); Ye et al. (2023), the ability for these methods to recover accurate human motion in the global frame leaves much to be desired.

GHMR is a much more challenging task than regular HMR, as we need to simultaneously constrain and predict the states of both major actors in HMR: 1.) the moving human and 2.) the camera capturing this moving human. Jointly optimizing the human-camera pair in predicting global motion requires great temporal understanding for both that allows the model to not just reason about the plausibility of its predictions on a per-frame basis, but rather the plausibility of the sequential progression of predictions across time. The lack of temporal understanding of the human-camera pair could yield a multitude of failure modes: for example, failing to ensure consistency between pose transitions and its corresponding global translation, hence resulting in highly inaccurate global root trajectory as well as foot sliding, and wrongfully

attributing the camera's jitters to the human, hence forcing the predicted human to jitter instead Ye et al. (2023); Li et al. (2022).

We propose a novel monocular GHMR framework that systematically optimizes both the human motion and camera movement with enhanced temporal understanding to recover a more accurate and plausible global human motion. More specifically, we introduce a framework that represents global human motion through a neural motion field Wang et al. (2022) supervised by a motion diffusion model (MDM) Tevet et al. (2022) serving as a motion prior and dynamic camera predictions initialized by DROID-SLAM Teed & Deng (2022).

MDM is used to constrain the predicted motion by penalizing implausible pose sequences outputted by the neural motion field. This motion prior is crucial as it leverages its knowledge on the inherent characteristics of human motion learned through training with a large-scale 3D human motion dataset et al. (2019), and thus possesses a strong prior for coherent human motion. Our multi-stage optimization framework ensures that the motion prior instills temporal understanding for the human and the dynamic camera without wrongfully tangling the two. We hereby refer to this motion diffusion-guided GHMR framework as **DiffOpt**.

We validate **DiffOpt**'s GHMR capability through evaluating its performance on videos from the Electromagnetic Database of Global 3D Human Pose and Shape in the Wild (EMDB) dataset Kaufmann et al. (2023) and comparing its performance alongside four other GHMR algorithms: GLAMR Yuan et al. (2022), SLAHMR Ye et al. (2023), WHAM Shin et al. (2024), and TRACE Sun et al. (2023). We verify that **DiffOpt** demonstrates the best performance in recovering global motion.

To summarize, our contributions to the field of global HMR through this work are threefold:

- We present **DiffOpt**, a motion diffusion and neural motion field-based GHMR framework for single human monocular videos that jointly optimize human motion and camera through leveraging a motion prior module and dynamic camera prediction module.
- We incorporate a motion diffusion-based Tevet et al. (2022) motion prior to guide the motion field's pose and global trajectory predictions towards realistic and plausible motions. Also, we successfully guide the global trajectory predictions using dynamic camera parameters from DROID-SLAM Teed & Deng (2022), demonstrating that neural motion field-based models are capable of handling videos captured by moving cameras.
- We validate our framework on video sequences from the EMDB dataset Kaufmann et al. (2023) and Egobody dataset Zhang et al. (2022b) and demonstrate superior global motion recovery capability against state-of-the-art global HMR methods particularly on long videos.

## 2 Related Works

### 2.1 HMR methods

Our method, **DiffOpt**, is an optimization-based global HMR framework on monocular videos (Cho et al., 2022; Zhang et al., 2022a; Guan et al., 2021; Iqbal et al., 2021; Sengupta et al., 2021; Kanazawa et al., 2018a;b; 2019), but it could also be seen as a test-time-optimization system that fine-tunes predictions from off-the-shelf methods. NeMo Wang et al. (2022) is another neural motion field-based TTO framework that jointly optimizes multiple video instances of the same action, which is a loosened form of multi-view data. NeMo aims to tackle spatial ambiguities of monocular videos such as occlusions through leveraging information gained from videos of alternative viewpoints. NeMo fine-tunes predictions from VIBE Kocabas et al. (2020). VIBE, which stands for Video Inference for Human Body Pose and Shape Estimation, is a 3D HMR system for monocular video sequences. VIBE has a temporal module that allows the system to leverage the temporal information available in videos, which helps in achieving more accurate and consistent 3D pose and shape estimations. SMPLify Pavlakos et al. (2019a) fits a 3D body model parameterized by the the Skinned Multi-Person Linear (SMPL) Loper et al. (2015) model to the 2D body joints predicted by DeepCut Pishchulin et al., a CNN-based model. SMPLify Pavlakos et al. (2019a) makes predictions based on monocular images, but an extension to the framework, named SMPLify-X Pavlakos et al. (2019b) estimates consistent 3D human poses across video frames by taking into account the temporal sequence of images.

## 2.2 GHMR methods

Recently, GHMR have also garnered attention, due to their ability to recover the global motion of humans, thus allowing us to analyze motion beyond the camera frame. Global occlusion-aware human mesh recovery with dynamic cameras (GLAMR) Yuan et al. (2022) fine-tunes pose predictions from HyBrik Li et al. (2021; 2023) through the use generative modeling to combat occlusions and recover the global trajectory of the human subject. Simultaneous Localization And Human Mesh Recovery (SLAHMR) Ye et al. (2023) recovers the global root trajectory of multiple humans by fine-tuning tracklets from PHALP Rajasegaran et al. (2021) through leveraging the transitional motion prior HuMoR Rempe et al. (2021) and dynamic camera parameters from DROID-SLAM Teed & Deng (2022).

## 2.3 Human motion priors

Human motion priors can be used to guide and constrain the estimation of human poses and motions in order to make them more physically plausible and consistent with our knowledge of how humans move. Arnab et al. (2019) use 3D joint predictions to compute a temporal error term that pushes the predictions to mimic the smoothness of natural human motion. Zhang et al. (2021) use a motion smoothness prior by training an autoencoder on AMASS data et al. (2019) to learn a latent space of motion that could be deemed as smooth. Rempe et al. (2020) utilize regression techniques on body joints and the contact points between the foot and the ground obtained from the input video to carry out a trajectory optimization, which could also be seen as a physics-based prior. Rempe et al. (2021) presents the 3D Human Motion Model for Robust Pose Estimation (HuMoR), an expressive generative model implemented as a conditional variational autoencoder that models a probability distribution of pose transitions. HuMoR is presented as a generative model, but its potential to serve as a motion prior for optimization-based HMR methods is also explored. SLAHMR Ye et al. (2023), a state-of-the-art global HMR algorithm, leverages HuMoR as a motion to constrain its predicted motion.

# 3 Method

In this section, we formulate the task of 3D GHMR (Sec. 3.1), discuss the mechanism of our motion diffusion prior (Sec. 3.2), and delineate the multi-stage optimization of **DiffOpt** (Sec. 3.3).

## 3.1 Problem set-up

Our objective is to recover the 3D *global human motion*, i.e. root trajectory included, given a video captured by a dynamic camera. We follow the paradigm of model-based HMR which uses the SMPL Loper et al. (2015) body model, which we refer to as $f_m$. Given an input video with $T$ frames, the global human motion is then represented by a sequence of articulation (a.k.a joint angles) $\boldsymbol{\theta}_{1:T} \in \mathbb{R}^{24 \times 3 \times T}$, global root orientation $\boldsymbol{\varphi}_{1:T} \in \mathbb{R}^{3 \times T}$, along with root translations $\boldsymbol{x}_{1:T} \in \mathbb{R}^{3 \times T}$.

**DiffOpt** is an optimization-based HMR method Pavlakos et al. (2019a); Kocabas et al. (2020). Specifically, we build on three types of models: (i) a 3D HMR regression method (e.g. HMR2.0 Goel et al. (2023)) that outputs only the articulation $\tilde{\boldsymbol{\theta}}$ for each frame, (ii) a 2D keypoint detection method (e.g. ViTPose Xu et al. (2022)) that outputs 2D joint keypoints $\tilde{\boldsymbol{j}} \in \mathbb{R}^2$, and (iii) a SLAM Teed & Deng (2022) method that estimates the extrinsic and intrinsic camera parameters per-frame. We represent the final, optimized, human motion using the recently proposed neural motion (NeMo) field Wang et al. (2022) for additional smoothness over the sequence and to seamlessly incorporate various loss terms including the MDM-SDS loss described in section 3.2. In other words, instead of optimizing the global motion $\{\theta, \varphi, \mathbf{x}\}$ directly, the variables are now represented using multi-layer perceptrons (MLPs) $\{f_{\boldsymbol{\theta}}, f_{\boldsymbol{\varphi}}, f_{\boldsymbol{x}}\}$ respectively. The articulation at frame $t$ is produced by NeMo as $f_{\boldsymbol{\theta}}(t)$, and similarly for the root motion. The full system architecture for the global motion prediction pipeline is in Figure 1.

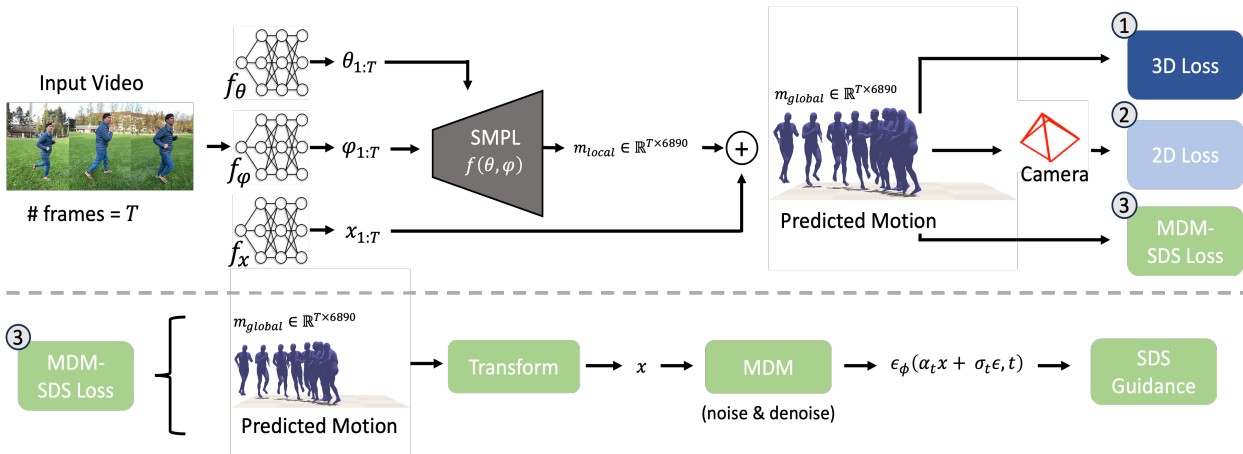

Figure 1: (top) **DiffOpt system architecture.** Given an input video with $T(n)$ frames, **DiffOpt** uses neural motion fields to predict the pose, root orientation, and global root translation for each frame. We regress these parameters using the SMPL Loper et al. (2015) body model to get the 3D joint and vertex positions. Our predicted motion is then constrained by 3D loss against initial predictions from off-the-shelf HMR models Goel et al. (2023), 2D re-projection loss against predictions from 2D keypoint detection models Xu et al. (2022), and motion prior loss from the motion diffusion model Tevet et al. (2022). (bottom) The MDM-SDS loss Poole et al. (2022) is computed by transforming the neural motion fields' predicted parameters to MDM's input format, running the noising and de-noising steps to compute the posterior, and using this to compute the SDS guidance Poole et al. (2022). This guidance term is back-propagated to the neural motion fields.

## 3.2 Motion Diffusion Prior

Motion priors Arnab et al. (2019); Rempe et al. (2021; 2020) is commonly utilized in the context of HMR optimization. In the context of GHMR with a dynamic camera, the human movement in the video is confounded by the movement of the camera. In addition, while one might expect that the camera movement can be well estimated using existing SLAM methods, and be factored out, we found that SLAM methods perform less robustly in these dynamic human-centric videos. Thus, utilizing motion prior is even more crucial in inferring the global root trajectory. **DiffOpt** utilizes the state-of-the-art motion prior, a motion diffusion model (MDM) Tevet et al. (2022). The key idea is to recover the accurate global human motion through leveraging MDM's strong prior of coherent human motion. To utilize MDM as a prior, we use the well-established technique of score distillation sampling (SDS) loss presented in DreamFusion Poole et al. (2022), which has been a major driving force in using image prior for 3D content generation.

*Motion diffusion model.* The core component of a MDM is a denoising network $\epsilon_\phi$ whose inputs are some noised motion and outputs are the denoised motion, denoted using $\mathbf{x}_0$. The forward Markov noising process follows:

$$q(\mathbf{x}_t|\mathbf{x}_0) = \mathcal{N}(\alpha_t\mathbf{x}_0, \sigma^2 I), \tag{1}$$

where $t$ is the diffusion timestep and $\alpha_t \in (0, 1)$ decrease monotonically. Commonly, the $\sigma$ is chosen to satisfy this constraint, $\alpha_t^2 = 1 - \sigma_t^2$. In other words, the MDM denoising network is trained with the following optimization:

$$\min_{\epsilon_\phi} \mathbb{E}_{\mathbf{x}_0 \sim \mathcal{D}, t \sim \mathcal{U}(0,1)}[\|\mathbf{x}_0 - \epsilon_\phi(\mathbf{x}_t, t)\|_2^2], \tag{2}$$

where $\mathcal{D}$ is a training set of real motion.

*Score distillation sampling.* Given a pretrained MDM, the SDS prior is formulated as:

$$\mathcal{L}_{\text{Diff}}(\phi, \boldsymbol{x}) = \mathbb{E}_{t,\epsilon}\left[w(t)\|\epsilon_\phi(\alpha_t\boldsymbol{x} + \sigma_t\epsilon, t) - \epsilon\|_2^2\right], \tag{3}$$

where $t \sim \mathcal{U}(0,1)$, $\epsilon \sim \mathcal{N}(0, I)$ and $w(t)$ is a weighting function (see Poole et al. (2022)). Lastly, to use the MDM-SDS prior in the HMR pipeline, the data representation has to match. Since MDMs are typically trained with auxiliary losses where the data includes the forward kinematic results (i.e. joint locations) and contact labels, we use the same (differentiable) transformation function on the HMR motion.

### 3.3 DiffOpt

We perform a 3-stage optimization, as delineated in Table 1. Intuitively, stage one warms up the neural field to mimic the articulation from the initial 3D HMR estimate from an off-the-shelf model Goel et al. (2023). In stage two, given the warmed-up articulation, we utilize the motion diffusion prior to complete a plausible global trajectory while updating the camera trajectory to keep the target human in view. In the final stage, we fine-tune both the human and camera motion using the 2D key-points using an ensemble of objectives.

#### 3.3.1 Stage 1: Articulation Warm-Up

In the warm-up stage, all 3 modules (pose, orientation, translation) are optimized through L2 loss with respect to initial predictions from HMR2.0 Goel et al. (2023). The warm-up optimization could be expressed as the following:

$$\min_{f_{\boldsymbol{\theta}}, f_{\boldsymbol{\varphi}}, f_{\boldsymbol{x}}} \mathcal{L}_{\text{warmup}}(f_{\boldsymbol{\theta}}, f_{\boldsymbol{\varphi}}, f_{\boldsymbol{x}}, \theta_{init}, \varphi_{init}, x_{init}) \tag{4}$$

$$\mathcal{L}_{\text{warmup}} = \frac{1}{T} \sum_{t=0}^{T-1} \left( \|f_{\boldsymbol{\theta}}(\tau_t) - \theta_{init}\|_2^2 \right.$$
$$+ \|f_{\boldsymbol{\varphi}}(\tau_t) - \varphi_{init}\|_2^2$$
$$\left. + \|f_{\boldsymbol{x}}(\tau_t) - x_{init}\|_2^2 \right), \tag{5}$$

where $\theta_{init}$ is the pose parameter, $\varphi_{init}$ is the orientation predicted by HMR2.0 Goel et al. (2023). The initial translation $x_{init}$ is estimated using a simple heuristic that keeps the human in the frustum of the estimated camera (see Supplementary Material). All three modules take $\tau_t$, an element at index $t$ of $\tau$, a self-normalized, monotonically increasing phase vector of length $T$ where $\tau_0 = 0$ and $\tau_{T-1} = 1$.

#### 3.3.2 Stage 2: MDM Guidance

The second stage, MDM guidance, is our key optimization step. Given the warmed-up articulation, our goal in this step is to first find a plausible global root trajectory coherent with the articulation. As we update the human trajectory, the rendered human will deviate from the original video. Hence, as we update the human trajectory, we also update the camera motion by making sure the reprojection loss stays low. Intuitively, this last step can be thought of as optimizing camera motion by human motion prior. In practice, we alternative between the steps of human motion update (Equation 6) and camera motion update (Equation 7).

*Learnable camera parameters.* In refining the camera motion, we learn four distinct parameters:

- **Camera Rotation Bias ($b_R \in \mathbb{R}^{6 \times T}$):** added to the camera rotation matrix in 6d rotation representation. Hence, the resulting camera rotation parameter is $R_{cam} = R_{SLAM} + b_R$, where $R_{SLAM}$ is the camera rotation predicted by DROID-SLAM Teed & Deng (2022).

- **Camera Translation Scale ($s_t \in \mathbb{R}^{1 \times T}$):** scales the camera translation vector.

- **Camera Translation Bias ($b_t \in \mathbb{R}^{3 \times T}$):** added to the scaled camera translation vector. Hence, the resulting camera translation parameter is $t_{cam} = t_{SLAM} * s_t + b_t$, where $t_{SLAM}$ is the camera translation predicted by DROID-SLAM Teed & Deng (2022).

- **Camera Focal Length Scale ($s_f \in \mathbb{R}^{1 \times T}$):** scales the focal length. Hence, the resulting camera focal length is $f_{cam} = f_{SLAM} * s_f$, where $f_{SLAM}$ is the focal length predicted by DROID-SLAM Teed & Deng (2022).

*Human motion update.* For optimizing human motion, we use a combination of the MDM-SDS loss and only the articulation loss from the warmup:

$$\min_{f_{\boldsymbol{\theta}}, f_{\boldsymbol{\varphi}}, f_{\boldsymbol{x}}} \left( \mathcal{L}_{\text{Diff}}(f_{\boldsymbol{\theta}}, f_{\boldsymbol{\varphi}}, f_{\boldsymbol{x}}) + \|f_{\boldsymbol{\theta}}(\tau_t) - \theta_{init}\|_2^2 \right), \tag{6}$$

*Camera motion update.* The optimization of camera motion can be written as:

$$\min_{b_R, s_t, b_t, s_f} \mathcal{L}_{2D} \quad , \text{ where} \tag{7}$$

$$\mathcal{L}_{2D} = \left( \frac{1}{T} \sum_{t=1}^{T} \rho(\boldsymbol{j}_t, \tilde{\boldsymbol{j}}_t) \right), \tag{8}$$

$$\boldsymbol{j}_t = P\Big( R_{cam} f_{3d}(\boldsymbol{p}_t) - \boldsymbol{t}_{cam} \Big), \tag{9}$$

$$\boldsymbol{p}_t = W\Big( f_{\boldsymbol{m}}\big(f_{\boldsymbol{\theta}}(\tau_t)\big) + f_{\boldsymbol{x}}(\tau_t) \Big). \tag{10}$$

We use $P$ to denote the perspective projection and $\rho(\cdot)$ the error function for 2D points. $W$ is a linear regressor fitted to get the major body joints in 3D through applying a linear transformation to the SMPL outputs. We use the Geman-McClure error function Barron (2019), which is more robust to outliers than the mean squared errors. $T$ indicates the length of the video.

| | Optimized params | Losses used |
|---|---|---|
| **1. Warm-up** | $f_\theta f_\varphi f_x$ | $\mathcal{L}_{\text{warmup}}$ |
| **2a. Human** | $f_\theta f_\varphi f_x$ | $\mathcal{L}_{\text{Diff}}$, $\|\theta_{\text{pred}} - \theta_{\text{init}}\|$ |
| **2b. Camera** | $b_R s_t b_t s_f$ | $\mathcal{L}_{2D}$ |
| **3. Fine-tuning** | $f_\theta f_\varphi f_x \; b_R s_t b_t s_f$ | $\mathcal{L}_{\text{Diff}}, \mathcal{L}_{2D}, \mathcal{L}_{\text{warmup}}$ |

Table 1: **Optimization parameters and loss functions used in different stages. DiffOpt** utilizes a multi-stage optimization scheme comprised of three distinct stages. In the warm-up stage, we optimize the neural motion fields with respect to initial predictions from off-the-shelf HMR methods. The second stage, named MDM-guidance step, is comprised of alternating between two sub-stages: 2a) Human optimization and 2b) Camera optimization stages. In the final fine-tuning stage, we optimize both human and camera with a compilation of losses.

### 3.3.3 Stage 3: Fine-tuning

In the final stage of **DiffOpt**'s optimization scheme, we fine-tune the human motion and camera motion jointly using $\mathcal{L}_{\text{Diff}}$, $\mathcal{L}_{\text{warmup}}$, and $\mathcal{L}_{2D}$. The fine-tuning optimization stage can be expressed as the following:

$$\min_{f_{\boldsymbol{\theta}}, f_{\boldsymbol{\varphi}}, f_{\boldsymbol{x}}, b_R, s_t, b_t, s_f} (\mathcal{L}_{\text{Diff}} + \mathcal{L}_{\text{warmup}} + \mathcal{L}_{2D}) \tag{11}$$

The neural motion field is constrained simultaneously by $\mathcal{L}_{\text{Diff}}$, $\mathcal{L}_{\text{warmup}}$, and $\mathcal{L}_{2D}$, and the dynamic camera parameters are constrained through $\mathcal{L}_{2D}$. Intuitively, the $\mathcal{L}_{\text{warmup}}$ serves to regularize the neural motion field

to prevent it from further deviating too much from the initial predictions from off-the-shelf predictors Goel et al. (2023), and $\mathcal{L}_{2D}$ is used to make fine-grained adjustments to our motion field for our predicted motion to better fit to pseudo-ground truth 2D keypoints Xu et al. (2022).

To summarize **DiffOpt**'s multi-stage optimization framework, stage 1 aims to initialize the neural motion field to mimic the initial predictions from HMR2.0 Goel et al. (2023), stage 2 grants the MDM module the ability to aggressively push the neural motion field to implicitly represent a more realistic and plausible motion when appropriate, and stage 3 finalizes our predicted motion through making minute adjustments for the motion field to have both the realism demanded by MDM and accuracy/faithfulness with respect to the original video sequence demanded by initial predictions and 2D keypoint supervision.

## 4    Experiments

In this section, we validate **DiffOpt**'s GHMR capability through recovering the global human motion in the EMDB dataset Kaufmann et al. (2023) and Egobody dataset Zhang et al. (2022b) video sequences. We conduct three primary GHMR experiments: first experiment evaluates **DiffOpt**'s performance on short 100-frame EMDB video sequences, second experiment assesses its robustness on lengthy, untrimmed EMDB sequences with varying lengths (averaging approximately 1,300 frames), and the final experiment evaluates on untrimmed Egobody sequences.

**Baselines**   To assess **DiffOpt**'s performance relative to pre-existing HMR methods, we compared it to the following baseline models:

- **GLAMR** Yuan et al. (2022) – a global HMR method that optimizes initial SMPL predictions from HybriK Li et al. (2021) and is robust to long-term occlusions and tracks human bodies outside the camera's field of view.
- **SLAHMR** Ye et al. (2023) – a global HMR method that optimizes initial SMPL predictions from PHALP Rajasegaran et al. (2021) and recovers the global trajectories of all humans in a moving camera video leveraging the HuMoR Rempe et al. (2021) motion prior and camera parameters from DROID-SLAM Teed & Deng (2022).
- **WHAM** Shin et al. (2024) – a global HMR method that learns to lift 2D keypoints to 3D mesh using motion capture data and video features to effectively integrate motion context and visual information.
- **TRACE** Sun et al. (2023) – a global HMR method that leverages a one-stage method to recover and track multiple 3D humans.

**Metrics**   We used four independent metrics to evaluate **DiffOpt** and the baselines.

- **MPJPE / MPVPE** – Mean per joint/vertex position error assess the accuracy of 3D HMR methods by determining the mean distance between predicted and actual joint or vertex positions in 3D space. Measured in millimeters (mm), MPJPE pertains to SMPL joints, whereas MPVPE considers SMPL vertices.
- **Global-MPJPE / MPVPE** – These metrics are global counterparts of MPJPE/MPVPE, factoring in predicted global root translation and orientation.

**EMDB Dataset**   We evaluate **DiffOpt** alongside the baselines with the EMDB dataset Kaufmann et al. (2023). The EMDB dataset includes 58 minutes of complex 3D human motion, totaling approximately 105,000 frames across 81 distinct sequences, captured in a variety of in-the-wild settings. We utilize the EMDB dataset slightly differently for our two experiments. For the first experiment involving 100 frame sequences, we evaluate **DiffOpt** on seven distinct sequences that contain motion that is characterized by both 1.) intricate and dynamic pose transitions and 2.) significant global root trajectory throughout the entire duration of the video sequence. The selected videos that suit both these criteria are: '09_outdoor_walk', '14_outdoor_climb', '16_outdoor_warmup', '32_outdoor_soccer_warmup_a', '37_outdoor_run_circle',

'41_indoor_jogging_workout', and '58_outdoor_parcours'. For simplicity, we refer to them as 'outdoor walk' 'outdoor climb', 'outdoor warmup', 'soccer warmup', 'outdoor run', 'indoor workout', and 'outdoor parcour' from this point and onwards. For each of the selected sequences, we divide each sequence to 100 frame segments. We do this because we observe that the performance of pre-existing baseline GHMR models degrade rapidly as the length of the input motion sequence increases. Hence, this experiment provides the baseline models the opportunity to demonstrate optimal performance. For the second experiment, we use the entirety of the EMDB dataset without trimming any sequences.

**Egobody Dataset** We also evaluate **DiffOpt** alongside a subset of baselines with the Egobody dataset Zhang et al. (2022b). Egobody is a large-scale dataset containing 125 distinct video sequences across 36 different scenes, and is intended to capture ground-truth 3D human motions during social interactions. We use the Egobody test set comprised of 17 distinct video sequences in an untrimmed manner.

## 4.1 Quantitative Results on EMDB

**DiffOpt** achieves the most robust overall performance compared to state-of-the-art baselines. Specifically, **DiffOpt** consistently outperforms most other methods in key global metrics (G-MPJPE & G-MPVPE), showing substantial improvements of 17% in G-MPJPE and 18% in G-MPVPE for trimmed sequences (more detailed discussion in section 4.1.1) from the third-best GHMR framework while marginally trailing behind WHAM's average global metrics. Note however, the average G-MPJPE and G-MPVPE of WHAM were computed excluding one sequence where WHAM optimization fails completely. More importantly, **DiffOpt** demonstrates 16% improvements in G-MPJPE and 16% in G-MPVPE compared to the second best framework for untrimmed sequences (more detailed discussion in section 4.1.2). Despite maintaining comparable performance in local metrics, **DiffOpt**'s superior global performance against long video sequences is crucial for applications requiring accurate global trajectory estimation. On the other hand, other methods that show smaller improvements in some local metrics have much less consistent and worse performance in some settings, in addition to the poorer global MPJPE / MPVPE performance in experiments on both trimmed and untrimmed EMDB sequences.

### 4.1.1 Trimmed EMDB Sequence Results:

We now offer a more in-depth discussion of the quantitative results of the first experiment shown in Table 2.

While the camera-frame MPJPE and MPVPE metrics are comparable for all five models, the G-MPJPE and G-MPVPE metrics indicate that **DiffOpt** outperforms GLAMR Yuan et al. (2022), SLAHMR Ye et al. (2023), and TRACE Sun et al. (2023) in terms of global human motion recovery. On average, WHAM Shin et al. (2024) achieves the best global metrics but fails completely for possibly the most difficult motion sequence of 'soccer warmup', hence demonstrating poor robustness even within this trimmed video setting. **DiffOpt** outperforms SLAHMR, GLAMR, and TRACE on 'outdoor climb', 'outdoor run', 'outdoor walk', and 'indoor workout', and also outperforms the aforementioned three baselines on average. **DiffOpt**'s superior ability to recover the global motion on the aforementioned sequences can be attributed to MDM's Tevet et al. (2022) ability to promote greater consistency between pose and global translation particularly in flat surfaces where the only external force is gravity. On the other hand, **DiffOpt**'s struggles in 'outdoor warmup' could be attributed to the fact that the human makes prolonged contact with rigid objects throughout the sequence. As MDM has been pre-trained on the AMASS dataset et al. (2019) comprised of motion where the human is only making contact with the ground plane, the sequence represents a challenging distribution shift.

### 4.1.2 Untrimmed EMDB Sequence Results:

On the second experiment, we run GLAMR, SLAHMR and **DiffOpt** on lengthy untrimmed EMDB sequences. We find that **DiffOpt** exhibits the best performance on the global metrics G-MPJPE and G-MPVPE, as shown in Table 4. Notably, **DiffOpt** scores 1776.2 in G-MPJPE, which is significantly better than GLAMR's score of 2113.5 and SLAHMR's score of 5595.8. This suggests that **DiffOpt**, compared to existing baselines, is the most robust GHMR framework against longer motion sequences. GLAMR yields

| Method | Outdoor Climb | Outdoor Warmup | Soccer Warmup | Outdoor Run | Outdoor Walk | Outdoor Parcour | Indoor Workout | Mean |
|---|---|---|---|---|---|---|---|---|
| | | | | MPJPE / G-MPJPE (mm) | | | | |
| GLAMR | 102.8 / 477.3 | 91.0 / 391.2 | 77.8 / 554.8 | 82.8 / 462.8 | 73.9 / 745.3 | 196.8 / 963.2 | 56.6 / 515.3 | 97.4 / 587.1 |
| SLAHMR | 69.5 / 247.6 | 95.5 / 299.7 | 77.4 / 345.7 | 76.6 / 169.4 | 64.6 / 425.4 | 90.3 / 697.4 | 57.0 / 536.8 | 75.8 / 388.9 |
| WHAM | 65.4 / 380.6 | 61.9 / 159.4 | NaN / NaN | 44.6 / 480.2 | 49.2 / 67.2 | 52.9 / 133.0 | 45.8 / 78.6 | 53.3 / 216.5 |
| TRACE | 97.4 / 676.9 | 79.9 / 378.2 | 72.4 / 1017.2 | 67.9 / 297.4 | 75.2 / 275.8 | 104.6 / 133.0 | 55.5 / 895.6 | 79.0 / 524.9 |
| DiffOpt | 90.7 / 241.4 | 94.0 / 364.0 | 74.1 / 357.8 | 82.6 / 130.8 | 82.0 / 288.4 | 82.5 / 662.5 | 91.6 / 213.4 | 85.4 / 322.6 |
| | | | | MPVPE / G-MPVPE (mm) | | | | |
| GLAMR | 124.2 / 496.8 | 116.0 / 439.4 | 98.0 / 553.4 | 106.9 / 479.2 | 93.0 / 687.3 | 270.1 / 962.6 | 73.1 / 501.5 | 125.9 / 588.6 |
| SLAHMR | 90.6 / 253.5 | 123.9 / 315.8 | 94.8 / 356.4 | 99.4 / 186.6 | 86.9 / 439.5 | 103.2 / 705.6 | 69.3 / 558.6 | 95.4 / 402.3 |
| WHAM | 79.2 / 392.4 | 83.4 / 183.6 | NaN / NaN | 55.2 / 467.1 | 63.9 / 79.4 | 68.6 / 141.3 | 63.0 / 82.5 | 68.9 / 224.4 |
| TRACE | 120.3 / 686.8 | 105.0 / 375.2 | 90.8 / 1041.4 | 81.5 / 300.4 | 99.3 / 352.5 | 120.8 / 141.3 | 71.0 / 897.5 | 98.4 / 542.2 |
| DiffOpt | 108.9 / 256.0 | 112.8 / 374.7 | 91.6 / 367.6 | 100.9 / 145.7 | 109.2 / 307.5 | 93.4 / 617.4 | 118.7 / 220.0 | 105.1 / 327.0 |

Table 2: **Quantitative results on the trimmed EMDB dataset.** We validate DiffOpt's GHMR capability by comparing its performance against GLAMR and SLAHMR on a subset of the EMDB dataset containing motions of outdoor climb, warmup, soccer warmup, outdoor run, walk, parcour, and indoor workout. While local MPJPE and MPVPE metrics are comparable across all methods, **DiffOpt** clearly stands superior in global metrics, with **DiffOpt** showing superior performance in five out of seven evaluation sequences. This result highlights **DiffOpt**'s robustness in recovering accurate global human motion, particularly in scenarios with significant global root trajectory.

the best local MPJPE and MPVPE metrics but trails against **DiffOpt** in its ability to recover global human motion. Lastly, SLAHMR optimization invariably breaks down when attempting to optimize the full untrimmed EMDB sequences, resulting in considerably worse metrics compared to GLAMR and **DiffOpt**. Validating **DiffOpt** to be the most robust GHMR method amongst state-of-the-art models holds an important implication that **DiffOpt** is the least constrained in terms of potential mocap applications scenarios.

| Method | MPJPE/G-MPJPE | MPVPE/G-MPVPE |
|---|---|---|
| GLAMR | 90.4 / 2113.5 | 114.1 / 2131.3 |
| SLAHMR | 234.8 / 5595.8 | 280.9 / 5596.6 |
| **DiffOpt** | 102.5 / 1776.2 | 130.2 / 1790.7 |

Table 3: **Metrics for original (untrimmed) EMDB sequences. DiffOpt** was evaluated alongside GLAMR and SLAHMR on original (untrimmed) EMDB sequences. **DiffOpt** achieves global metrics that are significantly better than both baselines, thus proving that **DiffOpt** is the most robust GHMR framework against lengthy sequences.

In conclusion, the quantitative metrics strongly indicate that **DiffOpt** outperforms existing state-of-the-art GHMR methods, particularly in global metrics, which is crucial for not only accurate global trajectory recovery but also recovery of motion with coherent root translation. **DiffOpt** not only demonstrates superior accuracy in these metrics across multiple challenging sequences but also maintains a significantly more robust optimization framework towards longer motion sequences than its closest competitor, SLAHMR Ye et al. (2023), as shown in table 4. These results underscore **DiffOpt**'s potential as the preferred solution for applications requiring robust GHMR in dynamic, real-world environments for prolonged motion sequences.

### 4.2 Quantitative Results on Egobody

**DiffOpt** achieves the best global HMR performance on the Egobody dataset comprised of 17 lengthy video sequences with an average of approximately 1,390 frames. More specifically, **DiffOpt** boasts an improvement of 24.6% in G-MPJPE metrics and 26.7% in G-MPVPE relative to the second-best method WHAM. SLAHMR's optimization framework encounters numerical instability and fails for all 17 sequences, hinting at

its poor robustness towards lengthy videos with over 100 frames. Moreover, we also evaluate **DiffOpt**after replacing our off-the-shelf DROID-SLAM camera predictions with masked-SLAM camera predictions proposed in TRAM Wang et al. (2024). In this revised setting, **DiffOpt** still demonstrates the best global HMR metrics relative to other state-of-the-art baselines.

| Method | MPJPE/G-MPJPE | MPVPE/G-MPVPE |
| --- | --- | --- |
| SLAHMR | NaN / NaN | NaN / NaN |
| WHAM | 94.1 / 572.7 | 112.1 / 596.9 |
| **DiffOpt**(TRAM cam) | 117.9 / 502.0 | 150.9 / 545.6 |
| **DiffOpt** | 129.2 / 459.8 | 160.6 / 471.1 |

Table 4: **Metrics for Egobody test sequences. DiffOpt** was evaluated alongside SLAHMR and WHAM on the Egobody test sequences. While SLAHMR's optimization framework completely fails to handle long video sequences and WHAM optimization shows poor robustness, **DiffOpt** achieves global metrics that are significantly better than both baselines, thus proving that **DiffOpt** is the most robust GHMR framework against lengthy sequences.

## 4.3 Qualitative Results on EMDB

We render **DiffOpt**'s estimated global human motion on both the original input video as well as a static world frame to assess whether **DiffOpt** has recovered human motion that is realistic, plausible, and faithful to the motion depicted in the video sequence.

On the original video renderings shown in Figure 2, **DiffOpt**'s human mesh perfectly encapsulates the human body at all time intervals, and the rendered limbs and feet position are faithful to the human's action. GLAMR renderings occasionally fail to fully cover the human body, while SLAHMR renderings have minor inaccuracies in feet position. The global root trajectory plots indicate that **DiffOpt** clearly recovers the most accurate global root trajectory. **DiffOpt**'s global root trajectory not only adheres to the ground-truth trajectory throughout the entire duration of the sequence but also has the most accurate final position, which indicates that **DiffOpt** most accurately estimates both the incremental translation along the way and the net translation.

Hence, the qualitative results demonstrate **DiffOpt**'s superior ability to recover realistic, plausible, and accurate human motion as well as global root trajectory for the whole duration of the motion.

In conclusion, the experimental results provide strong evidence that **DiffOpt** outperforms existing state-of-the-art GHMR methods both quantitatively and qualitatively. Quantitatively, DiffOpt achieves superior accuracy in key global metrics, consistently outperforming baselines across multiple challenging sequences in both EMDB untrimmed dataset and Egobody test set. This is particularly important for applications that require the precise global trajectory estimation for complex pose sequences. Qualitatively, **DiffOpt** demonstrates its ability to produce highly realistic and plausible human motion, with visually accurate and natural limb positioning and adherence to the ground-truth global trajectory. The rendered outputs show that **DiffOpt** effectively captures the intricate dynamics of human motion, even in sequences with complex movements.

## 4.4 Ablations

For ablation studies, we test the contribution of DiffOpt's two primary design components: 1. neural motion field, and 2. multi-stage optimization on all sequences of the trimmed EMDB dataset. We also include the metrics on the full **DiffOpt** model for comparison. The results of these ablation experiments are provided in table 5.

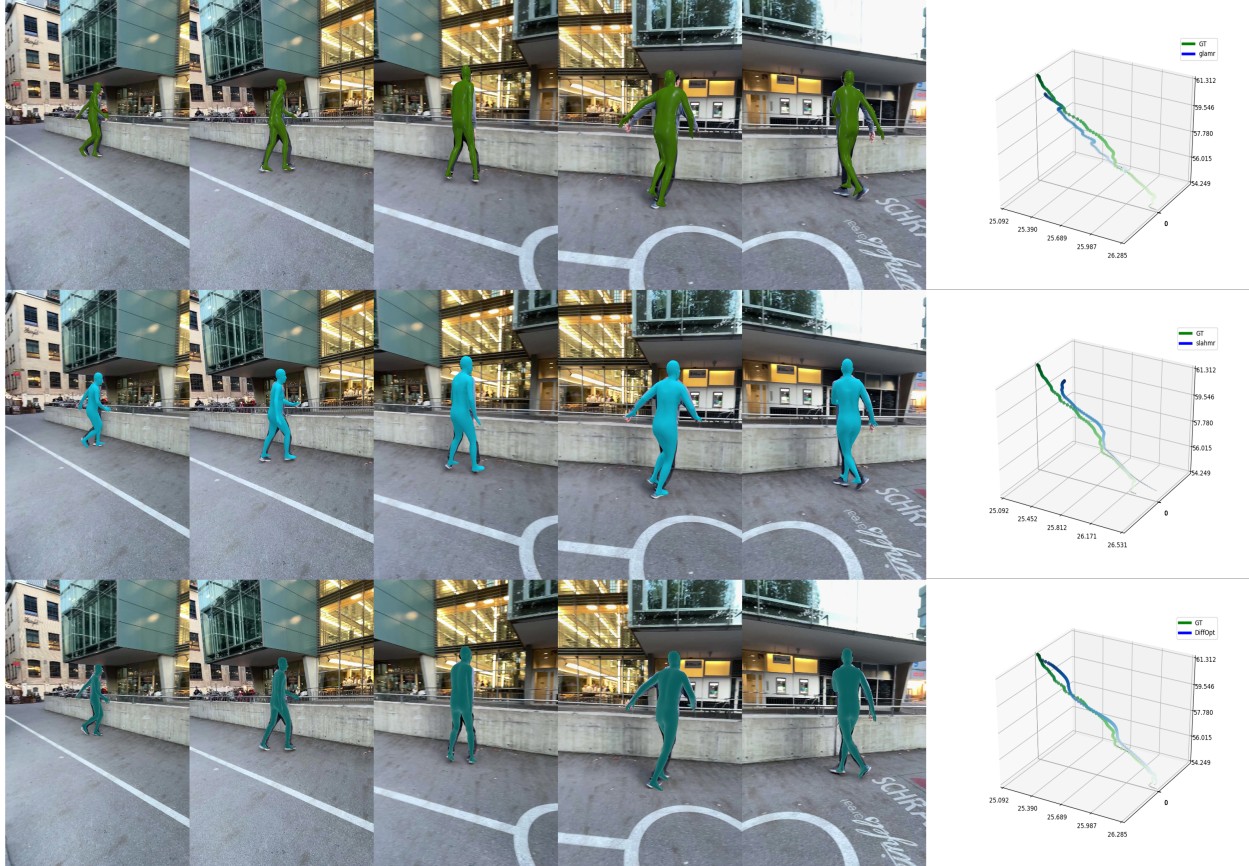

Figure 2: **Qualitative results on a trimmed segment in the 'soccer warmup' EMDB sequence Kaufmann et al. (2023)**. This is a challenging motion sequence, as the human subject continuously twists his hips while making quick side-steps. 3D human meshes have been rendered on the original video sequences for GLAMR Yuan et al. (2022) on the top row, SLAHMR Ye et al. (2023) on the middle row and **DiffOpt** on the bottom row. Moreover, the ground-truth global root trajectory and each model's predicted global root trajectory have been visualized next to the original video renderings.

In the ablation for neural motion field, we replace the motion representation with learnable parameters initialized with HMR2.0 values. Replacing the neural motion field MLPs with learnable tensors initialized with HMR2.0 values for pose, root orientation, and global translation maintained local MPJPE and MPVPE metrics but significantly worsened global metrics (G-MPJPE and G-MPVPE), highlighting the importance of implicit neural representations in the successful integration of the MDM motion prior for temporal consistency.

In the ablation for the multi-stage optimization scheme, we try two things: 1. replace the multi-stage scheme with a single-stage scheme that includes all loss terms, and 2. remove each of the three stages. For the first experiment, combining all loss terms into a single stage severely deteriorates performance, thereby demonstrating that the multi-stage optimization framework is essential for leveraging the MDM-SDS loss term. Next, bypassing each optimization stage independently revealed that the warm-up and MDM stage significantly impacts global metrics, and the fine-tuning step, though beneficial, is less critical.

## 5  Limitations and Future Work

Our current approach to 3D global human mesh recovery utilizing a motion diffusion model (MDM) has highlighted several areas for improvement that are crucial for enhancing the model's robustness and gener-

| Full Model Metrics | | | |
|---|---|---|---|
| | MPJPE | G-MPJPE | MPVPE | G-MPVPE |
| **DiffOpt** | 85.4 | 322.6 | 105.1 | 327.0 |
| Motion Representation | | | |
| Learnable params | 88.3 (+2.9) | 640.8 (+318.2) | 108.9 (+3.8) | 641.6 (+314.6) |
| Optimization Scheme | | | |
| Single Stage | 229.6 (+144.2) | 947.7 (+625.1) | 300.5 (+195.4) | 952.7 (+625.7) |
| Bypassed Stage | | | |
| No warm-up | 143.2 (+57.8) | 584.7 (+262.1) | 181.7 (+76.6) | 610.0 (+283.0) |
| No MDM step | 154.8 (+69.4) | 467.8 (+145.2) | 184.2 (+79.1) | 474.1 (+147.1) |
| No fine-tuning | 89.0 (+3.6) | 527.6 (+205.0) | 109.9 (+4.8) | 536.2 (+209.2) |

Table 5: **Ablation experiment results.** We explore the importance of **DiffOpt**'s implicit neural representation of motion and multi-stage optimization framework. The topmost row contains the metrics for the full **DiffOpt** model. Each ablation metrics are accompanied by its difference from the metrics of the full **DiffOpt** model in parenthesis.

alizability across various motion scenarios. We identified two main limitations within our MDM framework. Firstly, the model exhibits a decrease in performance when subjects maintain static leg postures over time, resulting in minimal translational movement of the human body. Secondly, the model struggles with interactions where the human body is subject to external forces beyond gravity and contact force from the ground, such as push-and-pull dynamics with external objects. Addressing these deficiencies is imperative for the model to reliably generalize across diverse scenarios.

## 6 Conclusion

We proposed **DiffOpt** a novel GHMR framework for recovering realistic and accurate global human motion given a monocular video captured under dynamic camera settings. **DiffOpt** jointly optimizes human motion and camera dynamics. It achieves this by integrating a motion diffusion-based prior with a dynamic camera prediction module in our multi-stage optimization scheme, which significantly improves the temporal coordination between the human subject and the camera.

We conduct extensive evaluations of our framework on the EMDB dataset, where it demonstrates enhanced capabilities in global motion recovery. Our method outperforms leading-edge global HMR techniques, including GLAMR and SLAHMR, showcasing its effectiveness in accurate human motion capture.

**DiffOpt**'s main contributions are not only in successfully integrating a motion diffusion model as a motion prior but also in proposing a multi-stage optimization scheme that enables the joint optimization of human and camera motion to disentangle the two motions and yield more realistic and accurate motions for each. Therefore, we believe that **DiffOpt**'s GHMR ability can continue to challenge the limits of GHMR model performance through seamlessly integrating better motion priors and camera parameter estimation algorithms into the optimization framework in the future.

**Broader Impact Statement**

**DiffOpt** offers several positive impacts by providing an accessible and cost-effective alternative to traditional marker-based motion capture systems, which require expensive equipment and specialized setups that are infeasible for everyday use. This heightened accessibility could benefit fields like sports science, healthcare, film, and gaming by enabling broader applications. However, there are potential negative impacts to consider: **DiffOpt**'s performance may vary across demographic groups if the training data lacks diversity, leading to biased outcomes in applications such as healthcare and sports analysis. Moreover, the method's reliance on initial predictions from existing pose estimation models and camera algorithms could also propagate any inherent biases or limitations in those models.

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

# A    Appendix

We have provided qualitative results in the form of body mesh rendering videos on several distinct EMDB sequences and can be viewed on our project page: https://sites.google.com/view/diffopt-tmlr

