# OpenReview forum: "Motion Diffusion-Guided 3D Global HMR from a Dynamic Camera"
_TMLR — Rejected by TMLR_

### Review · Reviewer_KqQW · 2024-09-24

**Summary Of Contributions:**

This paper proposes a novel 3D human mesh recovery model, DiffOpt, for tracking human motions across video frames. And the paper claims that the new method has better robustness and performance when compared with existing STOA HMR methods. The model consists of motion diffusion-based motion prior, a learned neural motion filed-based HMR model and a SLAM for camera pose estimation. The diffusion optimization process proposed here is separated into 3 stages: warm-up,  MDM guidance and fine-tune. And different loss for HMR, MDM-SDS loss and camera pose loss are used in these 3 stages. The model is tested on EMDB dataset with GLAMR and SLAHMR as baselines. Although the results on pre-frame metrics for all three models are similar, the global metrics for DiffOpt is better than the baselines. This supports the claim that DiffOpt has the most robust GHMR model.

**Audience:**

Yes

**Broader Impact Concerns:**

I think the currently broader impact statement is proper, clearly shows both the potential application of the method as well as the potential drawbacks and bottleneck for the learned model.

**Claims And Evidence:**

Yes

**Requested Changes:**

1. Include more datasets and baselines, given the multimodel output nature, dataset and baseline for SLAM and human 2D keypoint detection can be included as well.
2. Add more details about all the modules used in the model, including 2D keypoint detection, SLAM and 3D HMR regression.
3. Update the current model architecture image, to reflect how SMPL and MDM guidance work.
4. Address the performance gap between local and global metrics, and  justify why a model would be considered to be more robust if the per frame local performance is worse than baselines.
5. Add more detailed ablation study to justify the claimed effectiveness of multi-stage training.

**Strengths And Weaknesses:**

strengths
1. DiffOpt performance on EMDN is SOTA, both quantitatively on the metric and on the visualizations on their website. This shows the robustness and performance for the proposed method.
2. The idea of using pretrained motion diffusion prior for HMR although is not novel but is proven to be effective for a more robust HMR model. The idea of using score distillation sampling to utilize the MDM prior is interesting.
3. The ablation study is clear and useful. The result demonstrates the effectiveness of mutli-stage optimization and MDM motion prior.

Weaknesses
1. The lack of datasets and baselines. Currently the model is only tested on one dataset EMDB with two baselines, GLAMR and SLAHMR. Also EMDB is good dataset with diverse human motions, the model robustness across different environment and setting cannot be determined by one dataset with 81 distinct sequences.  Similarly although both baselines are SOTA, including more baselines would be ideal. And since the model is doing SLAM and human 2D keypoint estimation, dataset and baseline for those tasks can be used for testing.
2. The method part is unclear and hard to follow. Many important details of the model are omitted, for example, the 2D keypoint detection model is not motioned and 3D HMR regression method are not discussed, as well as the SLAM part. Although most of the modules are borrowed from existing works, I still think the paper should allow the reader to understand its core method without referring to other papers.
3. More discussion is needed to explain the gap between local and global metrices. Currently the authors claim that DiffOpt is the most robust HMR model based on its better global metrices compared to STOA. However the experiment section doesn't justify why the local metrices are lower than STOA and why lower local metrices can lead to better global metrics.
4, For the ablation study, the comparison of single stage and multistage optimization is not very convincing, depends on how differently losses are weighted and how the weights changes along training, single stage should have similar performance with multistage optimization. Current setup cannot support the claim of effectiveness of multistage training.

---

> ### Author Response · Authors · 2024-11-03
> **Response to Reviewer KqQW**
>
> Thank you very much for your feedback! In this response, we would like to address all the weaknesses and requested changes provided in your previous review.
>
> **1. Lack of datasets and baselines:** To address your concerns of the current work’s lack of diverse datasets and baseline methodologies, we provide two additional experiments:
> - Egobody test set evaluation of our methodology DiffOpt versus WHAM versus SLAHMR
> - trimmed EMDB dataset evaluation with two additional SOTA baseline global HMR models: WHAM and TRACE
>
> The metrics and discussion of these additional experiments have been provided in the response for all reviewers and can also be found in our revised manuscript's tables 4 & 2, respectively.
>
> **2. Method part is unclear:** We have intentionally decided to dedicate the vast majority of our discussion in our Methods section to the two major novelties of our framework DiffOpt:
> - MDM-SDS loss as a motion prior
> - multi-stage optimization framework
>
> Other aspects of our framework that you’ve mentioned (ex. 2D keypoint estimator, SLAM) are highly common off-the-shelf modules for most 3D HMR frameworks that we do not optimize or innovate upon, hence, we have limited our discussion of these submodules to ensure we are able to dedicate more of our paper to our proposed innovations and qualitative/quantitative results within the page limit.
> However, we would be delighted to provide a brief explanation for each of our off-the-shelf submodules:
> - ViTPose (2D keypoint estimator): Vision Transformer-based pose estimation model that detects human joint locations in 2D image coordinates. It uses a transformer encoder to process image patches and predict keypoint heatmaps using a simple convolution-based decoder.
> - HMR2.0 (3D HMR regressor): Regression-based HMR method that uses a vision-transformer encoder and a transformer decoder with a single “SMPL” token processed through cross-attention with transformer encoder output features.
> - DROID-SLAM (camera estimator): A deep learning-based simultaneous localization and mapping (SLAM) system that estimates camera motion and scene structure from video sequences. DROID-SLAM features an end-to-end differentiable architecture and iteratively updates camera poses and depth using the Dense Bundle Adjustment (DBA) layer.
>
> **3. Update the current model architecture figure for better portrayal of SMPL and MDM-guidance:** Thank you very much for this suggestion! We recognize that our current system architecture figure could be vague in terms of portraying the exact input & output of the SMPL body model, and how exactly MDM-guidance is used as one of our loss terms. We have made the necessary changes to address this concern in our revised manuscript! Please refer to Figure 1.
>
> **4. Performance gap between local and global metrics:** Thank you for this insightful question. We believe this question is a very important question to address because it alludes to the core contribution of our framework DiffOpt. The main problem DiffOpt attempts to solve is the problem of recovering accurate and realistic global human motion given a video data (temporal data) of a moving human captured by a moving camera. Local HMR, however, is a very different problem where the objective is to estimate only the pose of the human given the current frame. Thus, local HMR does not incorporate estimations of the global human orientation nor global root translation. The performance gap between DiffOpt’s local and global metrics can be summarized by this simple observation: DiffOpt demonstrates SOTA performance in global metrics but often lags behind other models in local metrics. And the reason behind this gap is simple: DiffOpt’s optimization framework is primarily focused on enhancing the global trajectory of the moving human through the MDM motion prior, so local HMR metrics are bottlenecked by local HMR parameters estimated by our off-the-shelf 3D HMR regression model (HMR2.0). Thus, the gap between local and global metrics for DiffOpt does not imply that our framework is less robust - in fact, experiments with longer whole video sequences indicate that DiffOpt is by far the most robust global HMR framework amongst other SOTA methods. An easy way to address the local-global metric gap is to simply swap out our current HMR regression model to a better off-the-shelf HMR model with better per-frame local HMR metrics and freeze the portion of our neural motion field responsible for pose prediction. Once again, however, this is far from the focus of DiffOpt.
>
> **5. More ablation to justify multi-step optimization:** We have provided a revised, more comprehensive ablation experiment results that utilize not just one specific sequence, but all of the trimmed EMDB dataset. Please refer to Table 5 on our manuscript.

---

### Review · Reviewer_t6ik · 2024-10-20

**Summary Of Contributions:**

- This paper presents a framework for human mesh recovery and camera pose estimation, incorporating a diffusion motion prior and neural fields into the human SLAM task.
- Extensive evaluation is conducted on benchmark datasets and compared to baseline HMR methods.

**Audience:**

Yes

**Broader Impact Concerns:**

No.

**Claims And Evidence:**

Yes

**Requested Changes:**

- Include a discussion on the potential scalability of the method to handle multiple humans.
- Discuss the differences between the proposed method and SynCHMR.
- Provide an evaluation on the EgoBody dataset.

**Strengths And Weaknesses:**

**Strengths**
- A practical framework that performs well on real-world monocular videos.
- Introducing diffusion motion SDS into HMR is a promising and interesting direction.
- The paper is well-written and easy to follow.

**Weaknesses**
- While methods like GLAMR and SLAHMR can handle multi-person videos, the proposed method is only applicable to single-person sequences.
- The proposed approach relies on hand-designed, multi-stage optimization, performing different optimizations on different parameters, which makes it appear more like a system-engineering effort.
- SynCHMR is a related method but is not mentioned in the paper.
- The evaluation dataset size is limited; an evaluation on the EgoBody dataset would be beneficial.
- Based on the results in Table 2, it’s unclear if the method provides a significant advantage over existing baselines.

---

> ### Author Response · Authors · 2024-11-03
> **Response to Reviewer t6ik**
>
> Thank you very much for your feedback! In this response, we would like to address all the weaknesses and requested changes provided in your previous review.
>
> **1. Restricted to single-human settings:** Though our optimization framework deals with optimizing the motion parameters of one moving human, it is not true that DiffOpt is only restricted to the single-human setting. DiffOpt can easily be adapted to be applicable in multi-human settings. As long as we possess initial SMPL predictions for multiple humans given the exact same video, we can iterate across all the humans and iteratively perform our optimization per-human. Thus, scaling our methodology to the multi-human setting is a simple software engineering task.
>
> **2. Relies on hand-designed multi-stage optimization framework:** Indeed, this is true! The core innovation of our optimization-based HMR is our carefully and deliberately crafted multi-stage optimization framework. Our intuition is simple: we leverage a pre-trained motion diffusion model (MDM) to constrain the moving human and allow the predicted motion to be plausible and “smooth” while we use 2D keypoint supervision to mainly constrain the camera (to be precise, the final fine-tuning step uses 2D keypoint supervision to make tiny adjustments to the human motion as well). The majority of optimization-based HMR methods also share this “multi-stage optimization” paradigm; GLAMR and SLAHMR both have multiple stages in their optimization pipeline, and more recent works such as WHAM and SynCHMR is also a multi-step optimization framework. Hence, we see the hand-designed multi-stage optimization framework as a strength rather than a weakness.
>
> **3. SynCHMR is not mentioned:** We would have benefitted from evaluating our framework DiffOpt alongside SynCHMR; however, SynCHMR has not released their codebase as of the date in which we are drafting this response. In terms of differences in methodology, there are several key differences that make DiffOpt fundamentally different from SynCHMR.
> - **Core focus/philosophy:** DiffOpt focuses on using motion diffusion as a motion prior for optimization. The key insight is leveraging MDM's strong prior of coherent human motion to guide the optimization process and correctly disentangle human and camera motions. On the other hand, SynCHMR focuses on marrying HMR and SLAM by using camera-frame HMR to help disambiguate SLAM. Thus, in terms of the big picture idea, DiffOpt's core approach to solving global 3D HMR is to understand the motion itself better, while SynCHMR's core approach is to understand the scene better.
> - **Use of diffusion-based models:** DiffOpt uses MDM as an explicit motion prior through score distillation sampling (SDS) loss, while SynCHMR uses a scene-aware SMPL denoiser that implicitly learns temporal consistencies and scene constraints.
> - **Scene understanding:** Once again, DiffOpt primarily focuses on human motion coherence and doesn't explicitly model scene geometry while SynCHMR explicitly reconstructs dense scene point clouds and uses them as constraints for human motion refinement.
>
> **4. Evaluation dataset is limited; add Egobody:** We have provided Egobody evaluations for DiffOpt alongside two SOTA global HMR methodologies WHAM and SLAHMR. The metrics and discussion of these additional experiments have been provided in the response for all reviewers, as well as on our revised manuscript. Please refer to Table 4 on the manuscript.
>
> **5. Unclear if the method provides significant advantages over other baselines:** Our revised experiments reinforce our claim that DiffOpt is the best global HMR method in long video settings. Since the primary objective of our framework is to tackle the problem of global human motion reconstruction, the two most important metrics to focus on are G-MPJPE and G-MPVPE. DiffOpt outperforms all SOTA baselines on both the original EMDB sequence and the Egobody test set in G-MPJPE and G-MPVPE metrics. Moreover, even in the highly constrained 100-frame video clip setting, DiffOpt achieves the second-best G-MPJPE and G-MPVPE values just behind WHAM.
>
> Moreover, in case your initial impression of unclearness mentioned in your review is from the gap between local and global metrics, we would like to kindly redirect you to the **4. Performance gap between local and global metrics** section in our"Response to Reviewer KqQW."

---

### Review · Reviewer_YPs1 · 2024-10-21

**Summary Of Contributions:**

The paper presents an optimization-based framework for global human pose estimation. Given an RGB video as input, the local SMPL pose parameters, 2D pose, and camera motion are estimated using off-the-shelf methods. The global human motion is then optimized in a stage-wise optimization framework in the form of neural motion fields adapted from an existing work. The parameters of the neural motion field are first optimized using the typical 3D and 2D joint losses. In the second stage, a motion diffusion model is used to apply the motion prior loss using score distillation sampling, which is the main contribution of the paper. Experiments are performed on the EMDB dataset where the proposed method is shown to outperform some of the existing methods - GLAMR (2022) and SLAHMR (2023).

**Audience:**

Yes

**Broader Impact Concerns:**

No ethical implications.

**Claims And Evidence:**

No

**Requested Changes:**

- Please compare results with WHAM and PACE as both papers were published at the time of the submission.
- Report the same evaluation protocol as followed by existing works.
- Ablation studies should be performed on the entire test set.
- It would be great to compare the contributions of the proposed method with PACE as the methods seem very similar i.e., multi-stage optimization frameworks with motion priors.
- Compare with motion priors other than VPoser i.e., NeMF used in PACE.
- Show qualitative results on longer videos.
- The recent method TRAM shows that the results of GHMR can be improved significantly by simply improving the camera motion. It raises the question if the proposed MDM-SDS loss would still yield similar improvements if the camera motion is already of high quality. This can be done by adopting the DROID-SLAM implementation from TRAM.

**Strengths And Weaknesses:**

# Strengths
- The paper is well-written and easy to read.
- Paper addresses a very challenging problem of global human pose reconstruction. A working solution would have numerous practical applications.
- The utilization of MDM-SDS loss makes sense and is shown to help.

# Weaknesses
**Lack of Novelty:**

The main weakness of the paper is the lack of novelty. The overall approach seems quite similar to PACE with the only addition of SDS loss using MDM which has become a pretty common practice in literature.

**Important related works are missing:**
- PACE 3DV'24
- WHAM CVPR'24
- TRAM ECCV'24 (not published at the time of submission, but available on arXiv for quite some time)
- COIN ECCV'24  (not available on arXiv at the time of submission, so I am just referring here for completeness). The method also uses the motion diffusion model as a prior.

PACE and WHAM both significantly outperform SLAHMR, while TRAM is currently the state-of-the-art method. Hence, it's important to compare these methods and highlight the contributions of the proposed method. Currently, the paper compares its results with rather old methods GLAMR (2022) and SLAHMR (2023).

# Experiments.
- It's unclear why the authors decided to follow a different evaluation protocol than the other published works. While the `Untrimmed` protocol makes sense, the trimmed version is similar to what existing methods do i.e., $MPJPE_{100}$, but with a subset of test videos. In any case, it's important to follow the original evaluation protocol to accurately compare the results with existing methods. The new protocols can be added additionally and should be motivated correctly.
- Results are only provided for the EMDB datasets. In general, the pose estimation community reports results on multiple datasets i.e., 3DPW, RICH, HCM, etc.
- Ablation studies are performed only on one sequence of the EMDB dataset, which is not sufficient to fully understand the contributions of different components. It's quite possible that the results do not hold on other sequences or datasets.
- The qualitative results in the supplementary material are shown for simpler sequences from EMDB. I would like to see the results of the `skateboarding` sequence as it truly reflects the benefits of joint optimization of human and camera motion. The authors mention in the limitation section that the current method doesn't work well in those cases, but it's not a valid reason to exclude them from evaluation.

Overall,  the paper lacks sufficient technical novelty and the experiments section is extremely weak. Hence the paper requires a *MAJOR REVISION*.

---

> ### Author Response · Authors · 2024-11-03
> **Response to Reviewer YPs1**
>
> Thank you very much for your feedback! In this response, we would like to address all the weaknesses and requested changes provided in your previous review.
>
> **1. Untrimmed vs Trimmed evaluation protocol:** Our separation into trimmed and untrimmed evaluations was deliberate and serves an important purpose in advancing the field. While we included trimmed evaluation (100-frame sequences) to provide fair comparison with existing methods like SLAHMR that are known to fail on longer sequences, our core focus and main contribution lies in the untrimmed evaluation using complete EMDB sequences. We believe this is critical as real-world applications rarely deal with perfectly trimmed 100-frame sequences - on a more general note, we believe evaluation protocols featuring such short and perfectly trimmed clips should be phased out to truly evaluate our models’ applicability in real-world settings. Our significant performance improvements on untrimmed EMDB sequences and strong results on the Egobody dataset outperforming all SOTA models demonstrate that DiffOpt is truly robust to lengthy, real-world videos. While comprehensive evaluation on trimmed sequences across the entire EMDB dataset would be ideal, we believe our current results on untrimmed data already effectively validate our key contribution: a GHMR framework that maintains accuracy even on challenging long-form videos, pushing the field beyond the artificial constraints of short, trimmed sequences. This robustness to sequence length is a direct result of our motion diffusion-based approach, which provides strong motion priors that remain coherent over extended temporal windows.
>
> **2. Results only provided for EMDB dataset:** The HPE community does indeed usually report results on multiple datasets like 3DPW, and RICH. However, all aforementioned datasets and the overwhelming majority of HMR datasets do not contain global trajectories, thus unsuitable for GHMR evaluation (ex. 3DPW’s ground-truth root trajectories aren’t in coherent world frames). We have provided additional evaluation of our framework alongside WHAM and SLAHMR on Egobody and show that DiffOpt outperforms all the baselines on G-MPJPE and G-MPVPE. Please refer to Table 4 on our revised manuscript!
>
> **3. Ablations:** We have provided a revised, more comprehensive ablation experiment results that utilize the entire trimmed EMDB dataset. Please refer to Table 5 on our revised manuscript!
>
> **4. Comparison with PACE:** While at first glance DiffOpt and PACE may look similar due to similar key phrases such as SLAM camera, multi-stage optimization, and motion prior, these two GHMR frameworks are vastly different.
> - **Core philosophy:** DiffOpt uses MDM as an explicit and powerful motion prior through score distillation sampling (SDS) loss. The key insight is that MDM contains strong priors of coherent human motion learned from extensive motion capture data. PACE, on the other hand, takes a system engineering approach focused on efficiency, using NeMF both as representation and motion prior.
> - **Multi-stage optimization:** DiffOpt employs a three-stage optimization focused on motion coherence through explicit MDM prior while PACE has a four-stage optimization emphasizing parallel processing efficiency.
> - **Experiments:** DiffOpt evaluates on complete, untrimmed sequences from real-world datasets (EMDB, Egobody). Hence, our framework deliberately tackles the harder problem of handling long sequences (averaging ~1,300 frames) and shows robustness to real-world challenges of lengthy videos. On the other hand, PACE evaluates their framework against their own synthetic dataset, and on Egobody, they evaluate merely on trimmed 100-frame clips.
> - **Camera optimization:** DiffOpt uses a more simple two-step approach where we first optimize the human motion with MDM guidance and then update camera parameters with reprojection errors to keep subjects in view. Hence, DiffOpt’s camera optimization is guided by MDM's understanding of plausible human motion, which implies that when camera estimates are poor, the strong motion prior from MDM still helps prevent implausible human poses. On the other hand, PACE directly integrates SLAM into bundle adjustment-inspired optimization and leverages a more complex optimization that involves pruning of human points from point cloud. To summarize, while PACE attempts to jointly optimize everything together with heavy reliance on background scene features, DiffOpt takes a more focused approach by first establishing plausible human motion through MDM's strong motion prior, then letting camera parameters follow this established motion - making it more robust in scenarios with unreliable background features.
>
> **5. Adopting better camera prediction like TRAM:** We have included an additional experiment where we swap out DROID-SLAM with TRAM’s “robustified” masked SLAM and demonstrate that in this setting, DiffOpt achieves state-of-the-art metrics in the Egobody test set.

---

### Author Response · Authors · 2024-11-03
**Overall response for all reviewers**

We would like to express our gratitude towards all reviewers for their insightful and constructive feedback! We seek to primarily address the common feedback amongst all our reviewers in this overall comment, which is: more baselines and more datasets for experiments. All three reviewers have commonly requested more baseline methodologies and datasets to be included in validating DiffOpt’s global HMR capabilities. Hence, we share 1.) Egobody evaluation with SLAHMR, WHAM, DiffOpt, and DiffOpt with a modified SLAM camera prediction introduced in TRAM, and 2.) trimmed EMDB evaluation with WHAM and TRACE as additional baselines. Inspired by many of your requested changes, we are delighted to share that the experiment section of our manuscript has been significantly improved through these additional experiments.

------

1.) Egobody test set evaluation with SLAHMR, WHAM, DiffOpt, and DiffOpt + masked SLAM camera (Table 4 on manuscript)

| Method             | MPJPE / G-MPJPE    | MPVPE / G-MPVPE    |
|--------------------|--------------------|---------------------|
| SLAHMR             | NaN / NaN          | NaN / NaN          |
| WHAM               | 94.1 / 572.7       | 112.1 / 596.9      |
| **DiffOpt**(TRAM cam) | 117.9 / 502.0   | 150.9 / 545.6      |
| **DiffOpt**        | 129.2 / 459.8      | 160.6 / 471.1      |

On the Egobody sequences, we fully demonstrate DiffOpt’s superior global HMR performance, as its G-MPJPE and G-MPVPE metrics are by far better than both SLAHMR and WHAM. Moreover, we also prove that even with a modified SLAM camera introduced in TRAM, another global HMR methodology, DiffOpt demonstrates global HMR capacities that are superior over all other baselines.

2.) Addition of WHAM and TRACE as baselines for trimmed EMDB evaluation (Table 2 on manuscript)

| Method   | Outdoor Climb   | Outdoor Warmup  | Soccer Warmup   | Outdoor Run     | Outdoor Walk    | Outdoor Parcour | Indoor Workout | Mean           |
|----------|------------------|-----------------|-----------------|-----------------|-----------------|-----------------|----------------|----------------|
| **MPJPE / G-MPJPE (mm)**    |                |                 |                 |                 |                 |                |                |                |
| GLAMR    | 102.8 / 477.3    | 91.0 / 391.2   | 77.8 / 554.8    | 82.8 / 462.8    | 73.9 / 745.3    | 196.8 / 963.2   | 56.6 / 515.3   | 97.4 / 587.1   |
| SLAHMR   | 69.5 / 247.6     | 95.5 / 299.7   | 77.4 / 345.7    | 76.6 / 169.4    | 64.6 / 425.4    | 90.3 / 697.4    | 57.0 / 536.8   | 85.5 / 388.9   |
| WHAM     | 65.4 / 380.6     | 61.9 / 159.4   | NaN / NaN       | 44.6 / 480.2    | 49.2 / 67.2     | 52.9 / 133.0    | 58.6 / 73.3    | 53.3 / 216.5   |
| TRACE    | 97.4 / 676.9     | 79.9 / 378.2   | 72.4 / 1017.2   | 67.9 / 297.4    | 75.2 / 275.8    | 104.6 / 133.0   | 55.5 / 895.6   | 79.0 / 524.9   |
| DiffOpt  | 90.7 / 241.4     | 94.0 / 364.0   | 74.1 / 357.8    | 82.6 / 138.0    | 82.0 / 288.4    | 85.2 / 662.5    | 91.6 / 213.4   | 85.4 / 322.6   |
| **MPVPE / G-MPVPE (mm)**     |                |                 |                 |                 |                 |                |                |                |
| GLAMR    | 124.2 / 496.8    | 116.0 / 439.4  | 98.0 / 553.4    | 106.9 / 479.2   | 93.0 / 687.3    | 270.1 / 962.6   | 73.1 / 501.5   | 125.9 / 588.6  |
| SLAHMR   | 90.6 / 253.5     | 123.9 / 318.5  | 94.8 / 356.4    | 99.4 / 186.6    | 69.8 / 439.5    | 103.2 / 705.6   | 69.3 / 558.6   | 95.4 / 402.3   |
| WHAM     | 79.2 / 392.4     | 83.4 / 183.6   | NaN / NaN       | 55.2 / 467.1    | 63.9 / 79.4     | 68.6 / 141.3    | 63.0 / 82.5    | 68.9 / 224.4   |
| TRACE    | 120.3 / 686.8    | 105.0 / 375.2  | 90.8 / 1041.4   | 81.5 / 300.4    | 99.3 / 352.5    | 120.8 / 141.3   | 71.0 / 897.5   | 98.4 / 542.2   |
| DiffOpt  | 108.9 / 256.0    | 112.8 / 374.7  | 91.6 / 367.6    | 100.9 / 145.7   | 109.2 / 307.5   | 93.4 / 617.4    | 118.7 / 220.0  | 105.1 / 327.0  |

The results on the trimmed EMDB sequence indicates that on short video clips lasting less than 4 seconds, DiffOpt’s performance slightly lags behind WHAM on average to be the 2nd best GHMR method, but demonstrates greater robustness than WHAM, as WHAM’s optimization fails completely for one of the seven sequences.

------

In accordance with our improved experimental findings, we would like to strongly emphasize our revised main claim: DiffOpt is currently the global HMR methodology that demonstrates the highest accuracy and robustness towards lengthy video sequences (beyond the highly restrictive setting of 100 frame sequences) and is therefore the least constrained in potential global motion capture applications. We have NOT claimed that DiffOpt is the current state-of-the-art global HMR framework in ALL possible settings, but our experiments have shown DiffOpt to outperform all baselines in long video settings, and outperforms all but one baseline in 100-frame clips.

---

### Decision · Action_Editor_EiiJ · 2024-12-20

**Recommendation:** Reject

**Comment:**

This has not been an easy decision to make for this paper. On one hand, there is strong evidence for the narrow claim, that will be of interest to a TMLR audience, but on the other hand, I wish there were a few more experiments to convincingly support the narrow claim. If I were to recommend acceptance, it would still be conditional on the following major revisions:

- That the authors try to get GLAMR and SLAHMR working on Egobody, or at least understand exactly why they fail
- Understand and explain why WHAM fails on one sequence on the EMDB dataset
- In the caption of Table 2 there is the following claim: "While local MPJPE and MPVPE metrics are comparable across all methods, DiffOpt clearly stands superior in global metrics, with DiffOpt showing superior performance in five out of seven evaluation sequences." I believe it is unfair to characterize the difference w.r.t. baselines on local metrics to be small enough to be "comparable", but use the word "superior" for the global metrics after excluding WHAM. I think the paper should have a better phrasing that avoids the use of the word "superior", which can be interpreted as SOTA.
- That all the remaining concerns of reviewer YPs1 are addressed (https://openreview.net/forum?id=6kX0hBNvGW&noteId=yF3zSARFOf)

Looking at the concerns of reviewer YPs1, I anticipate this list of fixes to be a major revision, so I am recommending that the paper be rejected for now. I strongly encourage the authors to revise and re-submit, as the paper is definitely interesting and is almost there in terms of having a complete set of experiments. So, please try again.

**Audience:**

This is of interest to an ML, vision, and even robotics and computer animation audiences.

**Claims And Evidence:**

The evidence provided by this paper has been challenging to evaluate. One of the main claims of the paper is that the proposed method "demonstrates superior global motion recovery capability against state-of-the-art global HMR methods particularly on long videos." In terms of global motion recovery metrics, the paper does not claim that it outperforms all baselines across all video sequences in the test datasets (EMDB and Egobody), and indeed it trails behind one of the baselines (WHAM) on one dataset (EMDB) but outperforms it on another dataset (Egobody). This makes drawing conclusions from this type of result challenging. So, the paper ends up making a narrower claim that applies to the long video sequences of the Egobody dataset, but not on EMDB: that the proposed method is best suited for long videos across all proposed methods. For this narrow claim, there is evidence in the paper to support it. However, there are instances of baselines failing completely on some test sequences (most notably SLAHMR on Egobody), and the paper does not make it clear what the source of these failures was and whether it was easy to fix or not. It was also not clear to me why GLAMR was not tested on Egobody. As such, it is difficult to say that there is conclusive evidence for the narrow claim. In terms of local motion recovery, the paper does not make any claims about outperforming existing baselines, and indeed performs worse on those metrics.

**Resubmission Of Major Revision:**

The authors may consider submitting a major revision at a later time.